# Connection between Osteoarthritis and Nitric Oxide: From Pathophysiology to Therapeutic Target

**DOI:** 10.3390/molecules28041683

**Published:** 2023-02-09

**Authors:** Huanyu Jiang, Piyao Ji, Xiaobin Shang, Yan Zhou

**Affiliations:** 1Department of Orthopedics, Renmin Hospital of Wuhan University, Wuhan 430060, China; 2Central Laboratory, Renmin Hospital of Wuhan University, Wuhan 430060, China

**Keywords:** nitric oxide, nitric oxide synthases, osteoarthritis, osteoclasts, osteoblasts, chondrocytes, inflammation

## Abstract

Osteoarthritis (OA), a disabling joint inflammatory disease, is characterized by the progressive destruction of cartilage, subchondral bone remodeling, and chronic synovitis. Due to the prolongation of the human lifespan, OA has become a serious public health problem that deserves wide attention. The development of OA is related to numerous factors. Among the factors, nitric oxide (NO) plays a key role in mediating this process. NO is a small gaseous molecule that is widely distributed in the human body, and its synthesis is dependent on NO synthase (NOS). NO plays an important role in various physiological processes such as the regulation of blood volume and nerve conduction. Notably, NO acts as a double-edged sword in inflammatory diseases. Recent studies have shown that NO and its redox derivatives might be closely related to both normal and pathophysiological joint conditions. They can play vital roles as normal bone cell-conditioning agents for osteoclasts, osteoblasts, and chondrocytes. Moreover, they can also induce cartilage catabolism and cell apoptosis. Based on different conditions, the NO/NOS system can act as an anti-inflammatory or pro-inflammatory agent for OA. This review summarizes the studies related to the effects of NO on all normal and OA joints as well as the possible new treatment strategies targeting the NO/NOS system.

## 1. Introduction

Nitric oxide (NO), an endogenous gas, is present in nearly all types of human cells. Initially, researchers did not recognize NO as a component of the human body but rather treated it as a new protein called “endothelium-dependent relaxing factor”. Currently, NO has been found to be a multipurpose intracellular signaling molecule. NO is used as a more sensitive substance for communication between the cells in different tissues and plays important roles in various physiological and pathophysiological responses, including blood pressure, blood circulation, platelet function, inflammation, neurotransmission, and immunity [1]. This review specifically focuses on the physio-pathological mechanism of NO in osteoarthritis (OA) and discusses the treatment potential of NO-related molecular structures for OA.

### 1.1. Epidemiology of OA

OA is the most common type of arthritis caused by cartilage degradation, which eventually leads to severe pain, limitation of activity, and impaired joint function. According to epidemiological studies, one in every eight people in the United States (27–31 million people among the total US population) is affected by symptomatic OA, while globally, OA affects 250 million people [2,3]. Furthermore, the prolongation of the average human lifespan [4,5] has led to increased OA prevalence and incidence. Recently, OA has become a major cause of impaired mobility and disability, thereby impairing the quality of life and increasing the economic and social disease burden.

### 1.2. Correlation between OA and NO

OA is caused by abnormal joint loads and mechanical stimulation, leading to a biomechanical process of abrasion [6]. Pathologically, the development of OA depends on chronic inflammatory processes, including immune cell infiltration and the release of cytokines and metalloproteinases into the joints [7,8]. NO is found in the synovial fluid, serum albumin, and urine of OA patients in the form of NO-derived molecules such as sodium nitrite, fluorophenyltryptophan, and N-methyl-L-arginine. The elevated levels of these molecules demonstrate the occurrence of OA, thereby showing the role of NO in bone and joint diseases [9]. The role of NO is essential for the development of OA. Therefore, studying the NO–OA interaction might deepen the comprehensive understanding of OA pathogenesis and provide guidance on OA treatment strategies.

## 2. NO and NO Synthase (NOS)

NO is an unstable, uncharged free radical, and the N atom in NO with free electrons gets rapidly oxidized. Hence, its biological lifetime is only several seconds; subsequently, NO is active only in the immediate proximity of NO-producing cells. Meanwhile, NO has a small size and favorable lipid solubility, thereby exhibiting high membrane permeability [10]. Therefore, NO can rapidly diffuse out of cells and cross through more than several microns into target cells, acting as a paracrine-signaling molecule.

NOS produces NO as a byproduct while oxidizing L-arginine to L-citrulline, using oxygen and nicotinamide adenine dinucleotide phosphate (NADPH) as substrates (Figure 1). In addition, this reaction requires flavin adenine dinucleotide (FAD), flavin mononucleotide (FMN), and (6R-)5,6,7,8-tetrahydro-L-biopterin (BH_4_) as cofactors. Among these cofactors, FAD and FMN are responsible for the transfer of electrons from NADPH to heme, and BH_4_ can bind to the oxygenase domain of NOS as an essential cofactor for substrates. Notably, this reaction is stereospecific; NOS can metabolize L-arginine, but not D-arginine [11]. There are three subtypes of NOS present in the body: neuronal NOS (nNOS; NOS1), inducible NOS (iNOS; NOS2), and endothelial NOS (eNOS; NOS3). They have the same NO production process; however, their structures and functions are tailored to the different body parts. nNOS and eNOS are constitutively expressed (thus also called constitutive NOS, cNOS) and can rapidly produce NO based on an increase in cytoplasmatic calcium. In some specific cases, such as oxidation of BH_4_ or depletion of L-arginine, eNOS can transform from a NO-producing enzyme into an enzyme that generates superoxide anion (O^2−^), which is called NOS uncoupling. iNOS is only induced when macrophages are stimulated by cytokines including tumor necrosis factor-alpha (TNF-α) and interferon-gamma (IFN-γ) and other factors such as bacterial toxoids. iNOS produces far more NO as compared to that produced by nNOS or eNOS [12]; this increased production might have various pathological effects.

NO is a reactive molecule that can act via numerous pathways, depending on the relative concentrations of NO and the surrounding environment in which NO is produced. The effects of NO can be direct or indirect, depending on the NO concentration—NO can show direct effects (<1 µM NO) as well as indirect effects (>1 µM NO). The indirect effects are induced by the reactive nitrogen species, produced by the interaction of NO with superoxide or oxygen [13].

In inflammatory diseases, NO acts as a double-edged sword. It can produce anti-inflammatory effects under physiological conditions. The agents, such as cytokines, promote iNOS activity, resulting in the synthesis of NO in large amounts by monocytes, macrophages, and granulocytes, among many other cells. NO then scavenges free radicals or kills microbes, thereby preventing cell injury.

NO can also act as a pro-inflammatory mediator. Its overproduction can be cytotoxic [14]. NO can react with superoxide and generate peroxynitrite. The subsequent secondary chain reaction leads to the production of NO_2_ and hydroxide, which can be even more toxic. This results in damaging the normal tissues, thereby increasing inflammation.

## 3. Physiological Effects of NO on Cartilage

As a chronic evolutionary disease, OA causes structural changes in the normal articular cavity. In order to comprehensively understand the mechanism of OA, the regulation of osseous tissue under physiological conditions should be explored.

The musculoskeletal system is an essential part of the human body’s functions and is affected by the biomechanical environment. The osseous tissue’s shape and density alter with the changes in the biomechanical environment. Qualitatively, the bone mass and density increase in areas with higher loads and decrease in areas with lower loads; this process is called bone remodeling. Bone remodeling consists of two steps: the stimulation of resorption in response to bone formation and the generation of new bone after bone degradation. Studies have shown the complexity of the process, involving multi-tiered communication networks of osteoclasts, osteoblasts, and other cell types in bone [15,16,17]. In addition, the coordination among endocrine, autocrine, and paracrine signals harmonizes human bone remodeling based on the adjustment of these cells in different stages [18]. Therefore, the recruitment, differentiation, and function of cells in bone remodeling are governed by a series of systemic regulators of bone metabolism (parathyroid hormone, vitamin D, and estrogen), local mediators (receptor of activated nuclear factor kappa-B ligand [RANKL], its antagonist osteoprotegerin, and Wnts/sclerostin), free radicals (superoxide, hydrogen peroxide, and NO), and bone matrix components [19]. The subsequent sections of this review article focus on the effects of the NO/NOS system on bone processes (Table 1 and Figure 2).

### 3.1. NO and Osteoclasts

Osteoclasts resorb bone and originate from somatic cells related to the single-core macrophage lineage [36]. Although they have unique bone resorption capabilities, they have various characteristics in common with macrophages, some of which are selectively expressed and regulated. This might be due to the adaptation of osteoclasts to osteoclast physiology, thus developing their specific functions. The similarities and nuances that distinguish osteoclasts from non-bone-resorbing macrophages reflect some specific mechanisms in bone remodeling. The NO/NOS system might be one such mechanism.

Thirty years ago, NO was confirmed to regulate the osteoclasts’ roles. Maclntyre reported that in isolated rat osteoclasts, NO could inhibit the spreading of the cells and bone resorption [37]. Moreover, nitrosyl-cobinamide has been recently reported as an immediate NO release agent, which reduces the number of osteoclasts in intact mice and inhibits the increase in osteoclast numbers in ovariectomized mice [20]. Amano found that during osteoclast formation in mouse bone marrow cells, the osteogenic helioxanthin derivative could inhibit the cyclic guanosine monophosphate (cGMP) degradation activity of phosphodiesterase, promote NO production, and inhibit the differentiation of osteoclasts dose-dependently [21]. Other experiments, such as the mouse skull assay and the rat long bone assay, also demonstrated similar results [38,39]. Taken together, these results indicated the inhibitory effects of high NO concentrations on bone resorption via two types of effects: an immediate inhibitory effect on mature osteoclast bone resorption and a reasonable inhibitory effect on precellular osteoclast differentiation. In this pathway, the effects of NO are cGMP-independent and different from the common pathway in most other systems. Notably, NO can induce mature osteoclasts to get rid of bone and reduce their acid secretion, thereby inhibiting bone resorption [40]. This process appears to be mediated by endogenous NO production and requires cGMP and protein kinase G (PKG).

In addition, lower NO concentrations can also promote osteoclast differentiation and survival. Interleukin (IL)-1β and TNF-α are powerful stimulators of bone resorption; they could appropriately enhance NO formation in the organ cultures of bone as well as bone marrow cultures. The addition of NOS inhibitors resulted in inhibiting the induced bone resorption, demonstrating that the NO/NOS system was stimulated by cytokines and other cytokine-induced mediators, such as prostacyclin (PG), to enhance bone resorption [39,41].

Kaneko et al. found that 8-nitro-cGMP, which is a NO derivate that is formed when NO reacts with cGMP in the presence of reactive oxygen species (ROS), could promote RANKL-induced osteoclast differentiation [22]. Moreover, a study assessed the levels of all three NOS modes in bones as well as isolated osteoblasts and osteoclasts using reverse transcription-polymerase chain reaction (RT-PCR) and immunohistochemistry [42]. Knocking out the nNOS gene increased the relative density of trabecular and cortical bone minerals in mice; the accurate measurement of the bone structure indicated a decrease in the total number of osteoclasts and osteoblasts, with worsened bone remodeling, which was reflected by the low mineral accumulation and bone formation rate [43,44,45]. These major manifestations indicated that nNOS might be essential for the differentiation and/or survival of all normal osteoclasts in vivo.

Moreover, nNOS-deficient bone marrow monocytes could produce poorly functional osteoclasts in vitro. Correspondingly, the iNOS-deficient mice showed no significant bone abnormalities, and their femoral length, trabecular bone volume fraction, bone formation rate, and osteoclast surface were normal [46]; this confirmed that the NO production by iNOS was much larger than that produced by nNOS, resulting in the loss of the promotion function. Therefore, in bone, iNOS might limit osteoclast activity by increasing NO levels, thus preventing excessive bone resorption in inflammatory diseases.

In other words, the high NO concentration can reduce the number of osteoclasts and inhibit their differentiation, demonstrating inhibitory effects on osteoclast bone resorption. On the other hand, the low NO concentration, mainly produced by cNOS, can promote the differentiation and survival of osteoclasts. Thus, the osteoclasts produced in nNOS-deficient individuals can lead to poor function. However, these conclusions are based on only in vivo and in vitro studies and lack the support of clinical data.

### 3.2. NO and Osteoblasts

Similar to their effects on osteoclasts, they might promote the differentiation and survival of osteoblasts and vice versa. The low NO concentration has been confirmed both in vivo and in vitro. Recent studies on human somatic cells and mouse models showed that the osteoblasts lacked argininosuccinate lyase, an enzyme participating in the synthesis of arginine and contributing to NO production, which resulted in low NO production and failure to differentiate [47]. Another study found that the vascular smooth muscle cells from the renal artery of male Wistar rats treated with aminoguanidine (AG; an iNOS inhibitor) showed a reduced expression of osteoblast differentiation factor (Cbfa1) [24]. Moreover, Wei et al. demonstrated that using the biochemical signaling molecules, such as PGs, NO, and insulin-like growth factor-1 (IGF-1) released by osteocytes, could increase osteogenesis, which showed a guiding significance in contemporary clinical treatment [48]. A study investigating the possible underlying mechanism of action showed that the proliferation of osteoblasts was simulated by cell-permeable cGMP analogs and prevented by pharmacological inhibition of soluble guanylyl cyclase (sGC) or protein kinase 2 (PKG2) or siRNA-mediated PKG2 knockdown; this suggests that the positive effects of low NO concentrations on osteoblast proliferation were mediated by sGC and PKG2 [25]. PKG2 exerts its antiapoptotic and proliferation-promoting effects in osteoblasts by activating Src, extracellularly regulated protein kinases-1/2 (Erk-1/2), and Akt [26]. The activated Akt phosphorylates and inactivates glycogen synthase kinase-3β, thereby stabilizing β-catenin and activating the Wnt pathway genes. The activation of the Wnt signaling pathway is a key factor in the differentiation, proliferation, and survival of (pre)osteoblasts and in driving bone formation [49,50]. In addition, a small dose of NO donors could activate the mRNA expression of osteoblast genes, such as alkaline phosphatase, osteocalcin, and collagen-1, and increase bone matrix synthesis and mineralization, thereby enhancing the osteogenic differentiation of (pre)osteoblasts in vitro [51,52,53].

In addition, based on estrogen stimulation, the moderate cNOS expression in osteoblasts can produce NO, which has a substantial role in the growth and development of osteoblasts as well as cytokine production [27,28].

The initiation of mechanical stimulation is highly important for the growth, development, and remodeling of bone [29,30]. When fluid flows through the bone canalicular system, the resulting shear stress stimulates osteoblasts and osteocytes to enhance their anabolism nonspecifically. With the increase in their proliferation and survival, the bone marrow stromal cells, osteoblasts, and osteocyte-like cells respond to fluid shear stress in vitro. This anabolic reaction requires moderate NO production from calcium-mediated eNOS.

The high NO concentrations resulting from NO donors or proinflammatory cytokines can effectively inhibit the growth and differentiation of osteoblasts [54,55]. These conditions often occur in inflammatory disorders and are related to the inhibitory effects of pro-inflammatory cytokines on bone formation. Based on the animal model of inflammation-mediated osteopenia, the active cytokines are the reason for the reduced osteogenesis [56].

In summary, the high production of NO by iNOS in an inflammatory environment can effectively inhibit the growth and differentiation of osteoblasts. In most physiological conditions, the appropriate NO concentrations can promote osteoblast function, which might be mediated by sGC and PKG2. The initiation of mechanical stimulation can also contribute to the proliferation and survival of osteoblasts through moderate NO production.

### 3.3. NO and Chondrocytes

The effects of NO on chondrocytes in vivo under normal physiological conditions are hard to observe due to the dominance of iNOS in these bone cells. In other words, a necessary stimulation is required for the chondrocytes to produce NO.

A study using the chick cartilage model reported that the NO metabolites played a role in the maturation and differentiation of chondrocytes [34,35]. All three NOS isoforms are expressed and remain active in the growth plate. At least two NO-mediated functions are important in epiphyseal chondrocytes. After the maturation of chondrocytes, NO and related compounds stimulate chondrocyte hypertrophy via the cGMP-dependent pathway, thereby increasing the expression of alkaline phosphatase and type X collagen (maturation markers). In the late stages, inhibiting NO production can inhibit apoptosis, while exposure to NO donors increases apoptosis, suggesting that NO induces this process. This suggested that based on artificial stimulation, moderate NO production might contribute to chondrocyte proliferation in vitro; however, relevant in vivo data are still lacking.

## 4. Pathological Effects of NO in OA

For a long time, OA pathophysiology was thought to be a process of long-term biomechanical wear caused by abnormal joint load and mechanical stimulation. Therefore, physical factors such as age, sex, obesity, and strain were frequently mentioned in the etiology [2,57]. Trauma was considered to promote this process, causing post-traumatic arthritis or secondary OA.

Recently, the importance of metabolic factors has also been reported. Both mechanical and biochemical factors are responsible for OA development [58]. The OA chondrocytes release various inflammatory mediators such as IL-1, TNF-α, and prostaglandin E2 (PGE_2_), upregulate iNOS and produce excessive NO, which leads to a perpetual release of inflammatory cytokines and other catabolic processes [33]. This affects numerous biomolecular processes in chondrocytes such as proteoglycan and collagen synthesis, as well as metalloproteinase and nuclear factor kappa-B (NF-κB) activation (Figure 3 and Table 2). The specific inhibition of iNOS results in decreasing the production of catabolic factors. In addition, the apoptosis of chondrocytes is also related to these processes [59].

### 4.1. Characteristics of OA Chondrocytes

Articular cartilage is a conjunctive tissue, consisting of only one type of cell (chondrocytes), encapsulated in a self-produced extracellular matrix (ECM) [60]. The ECM in cartilage contains water, collagen II, agglomerated proteoglycans, and hydrophilic biological macromolecules. It functions as a mechanical support and a lubricant for bones and joints.

The destruction of cartilage tissue structure in OA is related to changes in the ECM molecular components and can be divided into the following aspects: the formation of chondrocyte clusters, the presence of an irregular surface (fibrillations), the loss of cartilage volume, and matrix calcification [61]. The reduction in proteoglycans typically becomes more obvious with the progression of OA based on the results of optical biomarkers identified using Raman spectroscopy [62]. The distribution of collagen II also changes; it decreases in OA-degenerated areas and increases in chondrocyte cluster areas. The breakdown of ECM components is regulated by a set of aggrecanases such as A disintegrin and metalloproteinase with thrombospondin motifs (ADAMTS)-4 and -5 and collagenases such as matrix metalloproteases (MMPs), which are upregulated by NO [63,64,65]. In addition, the long-term increase in NO levels can significantly inhibit the release of gelatinase and PGs in chondrocytes [66,67]. In other words, the excessive release of NO might destroy the tissue structure of cartilage.

### 4.2. NO and Increased Matrix Degradation

NO promotes the reduction of the matrix via multiple pathways. NO plays a key role in the synthesis and degradation of proteoglycan and collagen in cartilage. NG-methyl-L-arginine (NMA) and thiocitrulline are potent NOS inhibitors, which can completely inhibit NO production. In a study using rabbit articular cartilage cultures, IL-1β and chondrocyte-activating factors were added to simulate the effects of OA in vitro by inhibiting proteoglycan synthesis and accelerating its breakdown. Both the NOS inhibitors substantially counteracted the suppression of proteoglycan synthesis [68]. The small-molecule inhibitors such as 3-(4-chloro-2-fluorophenyl)-6-(2,4-difluorophenyl)-2H-benzo[e][1,3]oxazine-2,4(3H)-dione (Cm-02) and 6-(2,4-difluorophenyl)-3-(3,4-difluorophenyl)-2H-benzo[e][1,3]oxazine-2,4(3H)-dione (Ck-02) could prevent TNF-α-mediated proteoglycan release into the culture supernatant of cartilage explants [69]. Moreover, luteolin could significantly inhibit IL-1β-induced NO production and reverse the degradation of collagen II [70]. Taken together, these results revealed that proteoglycan and collagen metabolism was related to NO.

NO can upregulate aggrecanases and collagenases. IL-1β could increase the mRNA expression levels of *MMP-1*, *MMP-3*, *MMP-9*, and *MMP-13* as well as some aggrecanases in mouse articular chondrocytes. Permine, an inhibitor of the NO/NOS system, could inhibit this upregulation, suggesting that NO was required for the activation of these mRNA expressions. MMP synthesis depends on the active cGMP, which is produced via the canonical signaling pathway of NO; this pathway is also a mechanism by which the increased amount of NO causes cell damage. Notably, the excessive NO concentrations (> 1 mM) can inactivate MMP-9 [71], which is different from the breakdown of the MMP–tissue inhibitor of metalloproteinases (TIMP-1) balance, suggesting that NO might have multiple functions in regulating MMP activity in OA.

In addition, the inhibitory effects of NO on matrix synthesis reflect the NO’s ability to affect the production and expression of autocrine growth factors, such as TGF-β1 and IGF-1 [72]. TGF-β1-neutralizing antibodies can completely block the repair effects of NMA on proteoglycan synthesis, suggesting the role of the TGF-β1 pathway in the ability of NMA to reverse these inhibitory effects. Meanwhile, the ability of IGF-1 to stimulate proteoglycan synthesis can also be antagonized by NO. This suggests that NO can inhibit matrix synthesis by interfering with the key autocrine and paracrine factors.

### 4.3. NO and Apoptosis

There is little proliferative activity in normal articular chondrocytes, and new cells are supplied from the ECM; therefore, the injury to chondrocytes and difficulty in repairing it are the central damaging features of OA. NO might be a key mediator of chondrocyte apoptosis [73]. Either endogenous or exogenous NO can induce apoptosis in chondrocytes via a mitochondria-dependent mechanism by activating caspase-3 and tyrosine kinases (Figure 4) [31,32]. In an in vitro model of fluid flow shear stress, Ren et al. found that an increase in NO production could reduce mitochondrial membrane potential and promote cytochrome C (Cyt C), another apoptosis mediator, to migrate from the mitochondria to the cytoplasm [74]. In their unilateral anterior crossbite in vivo rat models, the stimulator induced cartilage degeneration along with the upregulation of caspase-3 and caspase-9 expression, suggesting enhanced cell apoptosis. NOS inhibitors can block all these changes. Moreover, the expression of BCL2-associated X (Bax), Cyt C, and caspase-3 was upregulated in Sprague-Dawley rat models of sodium nitroprusside (SNP)-induced OA [75]. The treatment with carboxymethylated chitosan decreased the levels of apoptosis markers such as caspase-3 activation and DNA fragmentation as well as NO levels, suggesting caspase-dependent apoptosis. Another study reported that NO could inhibit autophagy and induce chondrocyte apoptosis [59]. Using SNP in chondrocytes could significantly reduce autophagic activity, autophagic flux, and the expression levels of several autophagy-related genes. This finding was supported by increases in ERK, Akt, and mammalian target of rapamycin (mTOR) phosphorylation. Moreover, the rapamycin-induced autophagy significantly suppressed NO-induced cell apoptosis via the mTOR/p70S6K pathway. In these processes, caspase-3 activation was only weakly detected, suggesting another caspase-independent apoptosis pathway.

Interestingly, Carlo and Loeser found that incubation with NO alone did not induce apoptosis in chondrocytes [76]. The diazeniumdiolates (NOC compounds), which are potent NO donors, did not cause cell death either. Further research found that the conventional treatment with compounds such as SNP and 3-morpholiosydnonimine (SIN-1), produced NO as well as ROS, demonstrating that both the NO and ROS were required to induce cell death [77]. In addition, the NOC compounds can shield against oxidative stress, likely by suppressing the chondrocyte energy metabolism, indicating that NO alone might have beneficial effects on chondrocytes.

### 4.4. NO and Inflammatory Mediators

As discussed in Section 3, OA chondrocytes spontaneously release various inflammatory mediators such as IL-1β, IL-6, IL-8, TNF-α, and PGE_2_, while normal chondrocytes do not release these mediators [78]. The induction process involves mechanical and biochemical factors. In general, the occurrence of abnormal mechanical stress causes biochemical and functional changes in cartilage tissues, which induce the production of these inflammatory mediators. Fermor et al. accurately measured the effects of static and intermittent compression physiological levels on NOS activity, NO production, and NOS antigen expression in porcine articular cartilage explants [79]. In the experiment, the static compression on the explant was 0.1 MPa for 24 h, and intermittent compression was 0.1 or 0.5 MPa at 0.5 Hz for 24 h. The results demonstrated both these compressions significantly improved the NO production and NOS activity. Notably, once the initial expression is activated, chondrocytes can release inflammatory mediators spontaneously, even without mechanical stimulation. There is a positive amplification loop in the cartilage tissues, maintaining its catabolic state. Even in the absence of abnormal biomechanical factors, IL-1 and other cytokines will keep on promoting the production of inflammatory molecules, such as NO. In addition, after its production, NO might promote IL-1 synthesis, probably through the mitogen-activated protein kinase (MAPK) and NF-κB signaling pathways [64,80], thereby creating a positive amplification loop that releases NO-mediated inflammatory cytokines. A study showed that NO could increase the synthesis of IL-18 and IL-1-converting enzyme (ICE) to some extent [81]. ICE is a caspase enzyme required for the maturation of IL-1β and IL-18. Feeding the animals with the specific iNOS inhibitor N-iminoethyl-L-dicalcium hydrogen phosphate (L-NIL) could significantly reduce the IL-18 and ICE levels in their femoral condyle and tibial plateau [82]. Besides, NO also participates in the reduction in the levels of trypsin retarder 9 (PI-9), a natural ICE inhibitor, thereby improving IL-1β and IL-18 activity [82].

Another significant cytokine-mediated effect is the activation of the NF-κB signaling pathway, which remains inactive in the cytoplasm due to the inhibitor of NF-κB (IκB) under normal physiological conditions. The stimulation with proinflammatory factors phosphorylates IκB, which then separates from NF-κB. The activated NF-κB migrates to the nucleus and facilitates the transcription of proinflammatory genes, such as IL-1β, TNF-α, iNOS, and other catabolic factors, including MMPs. In this whole process, NO is required for the continuous activation of the NF-κB signaling pathway. Taken together, these results revealed that the interaction between the NO/iNOS system and other inflammatory mediators might jointly lead to a series of pathological changes during OA development.

### 4.5. NO and ROS/Reactive Nitrogen Oxide Species (RNOS)

NO can combine with superoxide anions (O^2−^) and generate peroxynitrite, which can promote inflammation and cellular death in cartilage tissues. As compared to non-OA chondrocytes, the OA chondrocytes demonstrated an increase in IL-1-induced ROS. Blanco et al. found that ROS could promote chondrocytes toward necrosis, while NO just induced morphological apoptosis [83]. Both NO and ROS might cause DNA breaks. Chen used a comet assay to determine the level of DNA damage in non-OA and OA cartilage tissues treated with 0–500 mM NO donors (NOC-18 or SIN-1). The result demonstrated that, as compared to non-OA chondrocytes, the OA chondrocytes showed a significant increase in oxidative DNA damage (*p* < 0.01) [84]. An increase in the NO and ROS concentration is beneficial to increase DNA damage; the specific iNOS inhibitor N-(3-(aminohydroxy)benzyl) acetamidine (1400W) and superoxide scavengers such as superoxide dismutase (SOD) could inhibit this increase. Moreover, the combination of DNA damage and mitogenic stimulation can induce chondrocyte senescence [85]. Different from normal chondrocytes, OA cartilage chondrocytes have a proliferative characteristic, which might be due to the increased access of chondrocytes to proliferative factors in the synovial fluid due to fissuring or loosening of the collagen network or collagen matrix damage [86].

NO and peroxynitrite might have opposite effects on the activation of the NF-κB signaling pathway [87]. Clancy et al. found that incubating the chondrocytes of bovine cartilage tissue with IL-1β resulted in stimulating NF-κB in 40% of the cells, which was shown by the positive immunostaining of NF-κB subunits in the nucleus [87]. The treatment of cells with IL-1β and the NO donor S-nitrosocystine ethyl ester reduced the total number of chondrocytes with active NF-κB to 5%. Moreover, the incubation with IL-1β and peroxynitrite increased the proportion of chondrocytes with active NF-κB from 40% to 73%, indicating the reverse effects of NO. This suggested that NO might not be required for the immediate activation of NF-κB. Notably, the NO catabolic activity is partly mediated by peroxynitrite, and the increase in the NO levels in OA chondrocytes might be related to the elevated RNOS levels.

Peroxynitrite can also induce mitochondrial dysfunction in chondrocytes based on a calcium-dependent pathway, leading to caspase-independent cell apoptosis [31]. These studies once again show that NO and its derivatives fully exert multiple functions in the whole process of OA pathology.

### 4.6. NO and Pain in the OA Process

Constant and intense pain is a characteristic of severe OA, which can be considered an immediate reason for most patients to seek medical attention. Pain involves complex neural reflex processes [88]. Acute pain can be considered the body’s reaction to a potential injury, stimulating a response to limit such an injury. Pathologically, the release of inflammatory molecules such as IL-1β and IL-6 aggravates inflammatory reactions and cartilage degeneration and causes structural changes, such as osteophyte growth, meniscal injury, and synovitis; thus, they become a direct reason for pain [89]. In this process, NO causes a sustained release of inflammatory factors and is involved in the disease itself. Venkanna et al. conducted a study on S-methylisothiourea (SMT, an iNOS inhibitor) to investigate its role in relieving pain and inflammation. Wistar rats were orally administered with SMT at different concentrations daily after surgery. The mechanical hyperalgesia, thermal hyperalgesia, and tail flick latency after repeated flexion and extension of the OA knee were determined at weekly intervals. The results indicated that SMT could reduce mechanical hyperalgesia and serum levels of IL-1β, TNF-α, and nitrite in rats [90]. On the other hand, NO is also an important neurotransmitter. In persistent pain, NO can cause the central sensitization of the pain pathway, suggesting that nerves might remain sensitized even after the removal of stimulation. This is one of the processes involved in the change from acute pain to chronic pain. A study investigating the role of NO in inflammatory hyperalgesia indicated that the co-injection of PGE_2_ with an iNOS inhibitor, NMA, resulted in the inhibition of PGE_2_-induced hyperalgesia in rats, and the high levels of NO produced cGMP-dependent hyperalgesia [91].

Interestingly, recent studies have shown that NO can play a role in pain reduction. The low level of NO produced by cNOS might provide pain relief in OA, which might be related to blood flow, nerve-transmission pathway, opioid-receptor pathway, and anti-inflammatory pathway [92].

**Table 2 molecules-28-01683-t002:** Pathological effects of NO in OA and related mechanisms.

Pathological Effects of NO in OA	Related Mediators or Signaling Pathways	Related Mechanisms	Reference
Increases cartilage matrix degradation	Aggrecanases and collagenases	NO mediates upregulation of aggrecanases and collagenases	Wu et al., 2019 [63] Brown et al., 2020 [65]
Enhances chondrocyte apoptosis	Caspase-3 and -9, tyrosine kinases, Bax, Cyt C, and ROS	NO induces ROS generation and co-activates capase-3 to enhance chondrocyte apoptosis through a mitochondria-dependent mechanism	Poderoso et al., 2019 [31] He et al., 2020 [75]
Inhibits chondrocyte autophagy	ERK, Akt and mTOR signaling pathways	NO reduces autophagic activity, autophagic flux, and expression of several autophagy-related genes	Akaraphutiporn et al., 2020 [59]
Induces synthesis and release of inflammatory mediators	Inflammatory mediators and NF-κB signaling	NO interacts with inflammatory mediators which lead to constant inflammatory molecules release and NF-κB activation	Chen et al., 2008 [84] Moon et al., 2018 [64]
Promotes inflammation and cell death	ROS and RNOS	NO combines with O^2−^ to generate peroxynitrite which can cause DNA breaks and chondrocyte senescence, leading to cell necrosis	Copp et al., 2021 [85]
Cause central sensitization of the pain pathway	Inflammatory mediators and cGMP-dependent hyperalgesia	NO Promotes inflammation and structural changes of joint and play a role as neurotransmitter	Venkanna et al., 2014 [90] Aley et al., 1998 [91]

Overall, NO has various roles in OA development, which are far more complex than originally thought. In cartilage matrix, NO can promote matrix degradation as well as inhibition of matrix synthesis, which together cause damage to the matrix environment and thus involve the entire joint cavity. The NO-induced apoptosis further exacerbates the damage to the joint through caspase-dependent and -independent pathways; this condition can be hardly reversed due to the little proliferative activity of chondrocytes. Inflammatory mediators and the ROS/RNOS system are the main triggers of the inflammatory response. They interact with NO to form a positive amplification loop, leading to the continued release of inflammation-related molecules and causing further damage to the cartilage tissues. These factors complement each other, contribute to OA progression, and are eventually characterized by increased pain and limited mobility. This explains why early interventions for OA should be taken; with time, irreversible damage expands gradually. The specific measures are described in the next section.

## 5. NO in OA Treatment

The traditional OA treatment involves alleviating symptoms and relieving pain mainly using nonsteroidal anti-inflammatory drugs (NSAIDs) in combination with exercise and physiotherapy; sometimes opioids are also used in cases of severe pain [7]. However, the current medication has often limited effects [93] and the adverse effects make long-term treatment compliance difficult. For example, NSAIDS may lead to renal insufficiency, gastritis, and peptic ulcer formation [94,95]. Prolonged NSAIDs usage might also cause rare adverse effects, involving the cardiovascular and cerebrovascular systems [96,97,98]. Moreover, the end-stage OA relies heavily on artificial joint replacement, when the conservative treatments have had little effect [99]. Thus, we have been longing for a novel drug intervention as an early OA treatment. The NO/NOS system is an indispensable part of OA inflammation and might serve as a new target for its clinical treatment. The selective inhibition of iNOS, the major isoform expressed in the OA joint, can significantly decrease NO levels. Some important pharmacotherapeutic agents are listed in Table 3.

### 5.1. iNOS Inhibitors

iNOS inhibitors can weaken iNOS activity. Currently, there are four types of iNOS inhibitors (Figure 5 and Table 3), which are the most widely known and are classified based on the inhibitor-binding sites on the NOS [112]. The first type combines with the arginine site, parts of which are reaction-based since they require an active enzyme and NADPH for complete inhibition; most inhibitors belong to the first type. The second type includes a set of compounds that mimic the tetrahydrobiopterin cofactor. The third type interacts directly with heme, and some of them can bind to the heme of the enzyme monomer and prevent the formation of the active dimer; various anti-fungal imidazoles have been shown to inhibit NOS activity via this mechanism. The fourth type interacts with either calmodulin or flavin cofactors; however, this type of inhibitor is not considered a promising means for selective iNOS inhibition.

#### 5.1.1. N-Monomethyl-L-Arginine (L-NMMA)

L-NMMA is one of the first NOS inhibitors that has been widely used to decrease NO bioavailability. It is produced by the degradation of arginine-methylated proteins and exists naturally in living organisms. A study on simvastatin in rabbit articular chondrocytes showed that the L-NMMA treatment could abolish the expression of IL-1β-mediated cyclooxygenase-2 (COX-2) and NF-κB activation and enhance the effects of simvastatin on NF-κB activity, thereby reducing the damage caused by NO and ROS [100]. The percentage of iNOS inhibited by L-NMMA was >95%, leading to the complete inhibition of NO. Meanwhile, it could also increase PGE_2_ synthesis [113]. L-NMMA could reduce synovial inflammation and tissue damage in vivo. Because L-NMMA has no selectivity for the different subtypes of NOS, the inhibition of eNOS and nNOS might interrupt cellular communication, vascular tone, and neurotransmission. However, such results have not been reported yet.

#### 5.1.2. N-Iminoethyl-L-Lysine (L-NIL)

L-NIL is an arginine-based, moderately selective iNOS inhibitor. As compared to that for nNOS and eNOS, the iNOS selectivity of L-NIL is 23 and 49 times higher, respectively [114]. A lipid peroxidation (LPO) product, 4-hydroxynonenal (HNE), is considered an important mediator in the destruction of cartilage in OA. NO is a key inducer for triggering LPO based on peroxynitrite formation [115]. According to the experiments on inhibiting HNE synthesis in human OA chondrocytes and cartilage tissue explants by NO suppression, L-NIL could dose-dependently inhibit the IL-1β-induced NO release, iNOS activity, nitrated proteins, and HNE synthesis [103]. Moreover, L-NIL can also prevent the inactivation of HNE-metabolizing glutathione-s-transferase formation and inhibit the activation of p47 NADPH oxidase; all these are induced by IL-1β [116]. Furthermore, L-NIL can significantly reduce the makers of cellular death and apoptosis increased by exposure to the cytotoxic amounts of HNE as well as PGE_2_ and MMP-13 increased by the noncytotoxic amounts of HNE [103]. These findings showed that L-NIL could block the process of LPO and ROS production via NO-dependent and/or -independent pathways and reduce the HNE-induced cellular death, which might alleviate cartilage damage in OA.

#### 5.1.3. NG-nitro-L-Arginine Methyl Ester (L-NAME)

L-NAME is a non-selective NOS inhibitor, having a weaker inhibitory effect on NO production as compared to those of L-NMMA and L-NIL [117]. L-NAME could significantly decrease NO production and inhibit immune suppression during OA initiation in rats, along with decreased levels of MMPs and increased levels of TIMP, thereby protecting cartilage tissues from damage in the early stage of OA [118]. In addition, prophylactic L-NAME could reduce joint pain in rats, which were subjected to the anterior cruciate ligament transection of the right knee for experimental OA [104].

#### 5.1.4. 1400W

1400W is a highly selective iNOS inhibitor with high tissue and cell penetration abilities. It can increase the expression of Dickkopf-1 and frizzled-related proteins, which are attenuated by the IL-1β-induced upregulation of iNOS, which in turn activates the transcription of Wnt-pathway genes in human chondrocytes [119]. 1400W can also inhibit the cytokine-induced expression of MMPs and cellular apoptosis. However, its high dosage has toxic side effects, which might limit its application in patients.

#### 5.1.5. Cindunistat

Cindunistat hydrochloride maleate, also called SD-6010, is a human selective type II iNOS inhibitor that could reduce the production of synovial fluid nitrite as well as osteophyte formation in animal models. It can also relieve inflammation and pain. In a 2-year double-blind randomized controlled trial (RCT), cindunistat improved the radiographic features of Kellgren–Lawrence Grade 2 (KLG2) OA patients in the first 48 weeks; however, the improvement did not last after 96 weeks. Among the patients with KLG3, OA progression persisted, showing no improvement in either 48 or 96 weeks [106]. Meanwhile, a meta-analysis investigating the long-term RCTs of medications for knee OA suggested that cindunistat exhibited little effect [120]. Thus, cindunistat might have effects only on the early stage of OA. This is one of the few cases of NOS inhibitors being used in clinical trials, which revealed that with the progression of OA, inflammation damage cannot be easily reversed.

#### 5.1.6. Aminoguanidine (AG)

AG is an iNOS inhibitor that can inhibit oxidation, apoptosis, and inflammation. It has been reported to reduce the protein expression levels of iNOS and p65 in the liver of a diabetic animal model and reduce osteocyte apoptosis during non-traumatic osteonecrosis [121,122]. According to Ma et al., AG could reduce the gene expression levels of *iNOS* and *COX-2* in IL-1β-induced rat articular chondrocytes by suppressing the NF-κB activity and inhibiting the phosphorylation of IκBα and inhibitor of kappa B kinase-β (IKKβ), thereby showing its anti-inflammatory and cartilage protective effects [107].

### 5.2. Pharmaceuticals Inhibiting iNOS Expression

NO production can be inhibited by downregulating iNOS expression. Several types of drugs can attenuate iNOS expression through different mechanisms. NF-κΒ and signal transducer and activator of transcription-1 (STAT-1) are the key transcription factors for iNOS [123]. Inhibiting these signaling pathways might be conducive to reducing iNOS expression. Moreover, regulation at the post-transcriptional level also matters. By interfering with the amount of protein modifications, the balance of iNOS synthesis and degradation can be regulated.

#### 5.2.1. Glucocorticoids (GCs)

GCs are steroid hormones that are naturally synthesized in the human body or manufactured synthetically and possess anti-inflammatory effects. Hossein et al., indicated that dexamethasone (DEX; a type of GCs) could reduce the gene expression levels of *COX-2*, *iNOS*, *IL-1β*, *IL-18*, and *TNF-α* in synoviocyte as well as the production of NO and PGE_2_ in monocytes and macrophages, probably by inhibiting the NF-κB and destabilizing the *iNOS* mRNA [124]. Notably, using excessive GCs can cause serious, life-threatening side effects, thereby restricting their applications.

#### 5.2.2. Hyaluronic Acid (HA)

HA is synthesized by chondrocytes and fibroblasts in the human body as a component of the ECM in articular cartilage tissues. A study on mice subjected to collagen-induced arthritis demonstrated that, as compared to carnosine alone, the combination of HA and carnosine significantly reduced expression levels of iNOS as well as proinflammatory cytokines, chemokines, and COX-2 [125]. The possible mechanism of HA function involves the protection of protein kinase C alpha (PKCa), a factor related to chondrocyte proliferation, against NO inhibition—similar to a PKCa agonist used in articular cartilage.

#### 5.2.3. CoenzymeQ10 (CoQ10)

CoQ10 is a lipid that participates in the electron transport chain and aerobic respiration as a coenzyme for mitochondrial enzymes and exhibits potent antioxidant and anti-inflammatory abilities. CoQ10 could inhibit the NF-κB pathway and its downstream inflammatory mediators, including MMP-9, MMP-13, IL-1β, nitrotyrosine, and receptors for advanced glycation end products [101,126]. It could also significantly reduce the monosodium iodoacetate-induced upregulation of iNOS expression. Treatment with CoQ10 could considerably attenuate pain and cartilage degeneration in animal models [108].

#### 5.2.4. *Sargassum serratifolium*

*Sargassum* is a genus of marine brown algae (Phaeophyceae) and is widely distributed in the ocean. It has a highly edible and medicinal value. Current studies have shown that the extracts of *Sargassum serratifolium* have powerful anti-inflammatory and antioxidant effects [127,128]. Park et al. treated IL-1β-induced human chondrocytes with the ethanol extract of *Sargassum serratifolium* (EESS); the results showed a reduction in numerous inflammatory markers, including NO, MMPs, and ROS. Their results demonstrated that EESS could inhibit the activation of NF-κΒ, p38 MAPK, and PI3K/Akt signaling pathways, thereby inhibiting iNOS and COX-2 expression [109].

#### 5.2.5. Wogonin

Wogonin (5, 7-dihydroxy-8-methoxyflavone) is a naturally occurring flavonoid derived from the root extracts of *Scutellaria baicalensis*. According to Khan et al., Wogonin could completely suppress the expression of IL-6, PGE_2_, iNOS, and COX-2 and reduce the NO production in IL-1β-induced OA chondrocytes, thereby playing an anti-inflammatory role. Wogonin activated the ROS/ERK/nuclear factor erythroid 2-related factor 2 (Nrf2)/heme oxygenase-1 (HO-1)-SOD2-NADPH: Quinone Oxidoreductase-1(NQO1)-glutamate-cysteine ligase catalytic subunit (GCLC) signaling axis to induce low levels of ROS, thereby mediating the Nrf2/ARE pathways and suppressing the molecular events involved in oxidative stress and inflammation [110]. Moreover, wogonin also decreased the release of MMPs and ADAMT-4 and elevated the expression levels of cartilage anabolic factors, including collagen type II alpha 1 (COL2A1) and aggrecan (ACAN), to suppress matrix degradation [129,130].

#### 5.2.6. Myricitrin (Myr)

Myr, a flavonoid compound extracted from *Myrica rubra*, exhibits anti-inflammatory, antioxidant, and anti-fibrotic effects. Myr can inhibit the expression of iNOS and COX-2, thereby blocking NO and PEG_2_ production in IL-1β-stimulated OA rats [111,131]. Immunohistochemical analysis showed that the anti-inflammatory effects of Myr were based on the interference of IL-1β-activation of NF-κB and MAPK signaling pathways. The OA Research Society International scores were decreased in surgically induced mouse models in vivo, demonstrating the amelioration of OA development [111].

### 5.3. Applications of Materials Science

The traditional pharmacotherapeutic strategies have unavoidable flaws; the absorption process of the drug can be affected by various factors, and the patients’ compliance is difficult to guarantee. Therefore, researchers attempt to explore novel approaches in the field of materials science. Wang et al. used umbilical cord mesenchymal stem cells (UCMSCs), loaded with graphene oxide (GO) granular lubricant, for the treatment of knee OA in animal models [132]. In the animal group given the combination of USMSCs and GO, the NO contents decreased significantly in the serum and articular fluid. GO granular lubricant might promote the therapeutic effects of UCMSCs as scaffold-carrying material. Moreover, Shah et al. used growth factors important to chondrogenesis and encapsulated recombinant proteins in poly (lactic-coglycolic acid) 85:15 (PLGA) to fabricate a synthetic artificial stem cell (SASC) system. After injecting SASC, it dramatically reduced the NO contents and gene expression levels of ADAMTS5 and the proteoglycan 4 (PRG4) genes in the OA rodent model, thereby demonstrating significant anti-inflammatory and chondroprotective effects [133].

In another study, nanocomposites showed potent osteoinductive and free radical scavenging abilities. Nano-titanium dioxide (TiO_2_) with chitosan and chondroitin 4-sulfate (TCG) could reduce the formation of free radicals, such as NO, increase the total antioxidant activity, and upregulate the expression of osteoblast-inducing genes in the MG-63 cell line [134]. In addition, pharmacotherapy can combine with nanocomposites, forming nano-targeted drug delivery systems. By responding to specific internal stimuli (reduction-oxidation, pH, and enzymes) and external stimuli (temperature, ultrasound, magnetic, photo, voltage, and mechanical friction), nano-targeted drug delivery systems can deliver drugs at the proper concentration, time, and space, which overcomes the shortcomings of exceeding scope and premature leakage of traditional drug delivery systems. In comparison, this system greatly improves the efficiency of drug delivery and minimizes the side effects of drugs [135]. These results suggested that the nanocomposites might have a promising application prospect.

### 5.4. Physical Methods

In addition to the above strategies, some physical methods have also been reported to downregulate NO levels. For instance, hyperbaric oxygen (HBO) can inhibit iNOS expression and chondrocyte apoptosis in rabbit cartilage defects [136]. At higher oxygen tension, the NO, PGE_2_, and MMP activities decreased in the TNF-α-induced chondrocytes [137]. The combination of HBO with low-intensity pulsed ultrasound can increase matrix and TIMP production and inhibit iNOS expression and NO production [138]. Low-level laser therapy can reduce the expression levels of inflammatory markers, such as TNF-α, iNOS, and IL-1 [139]. Extracorporeal shock-wave therapy could improve pain and motor dysfunction in OA rabbits, in which NO level and chondrocyte apoptosis decreased significantly [140]. In general, these approaches require further clinical research.

## 6. Discussion and Future Perspectives

Current studies demonstrate that NO plays a critical role in activating various signaling pathways. Among these signaling pathways, NF-κB might be one of the most noteworthy. The activated NF-κB can promote the synthesis of proinflammatory factors, such as IL-1β, TNF-α, and iNOS, thereby aggravating the development of OA. In addition, MAPK, STAT-1, and PI3K/Akt signaling pathways have similar functions. Moreover, NO-activated ERK-Akt-mTOR signaling pathways can inhibit chondrocyte autophagy and promote apoptosis, and MMPs/TGF-β1/IGF-1 signaling pathways can increase matrix degradation, thereby causing cartilage tissue damage. On the other hand, the low NO concentrations can activate the RANKL signaling pathway, which produces downstream molecular 8-nitro-cGMP to promote osteoclast differentiation, as well as the Src-Erk-1/2-Akt signaling pathways, which can promote osteoblast proliferation and has anti-apoptotic effects, thereby enhancing bone remodeling.

Therefore, NO plays not only an inflammatory mediator role in OA but also a normal bone cell-conditioning agent. NO is necessary during the growth, differentiation, and maturation process of several types of bone cells. The pathological increase in NO contents due to stimulatory factors, such as abnormal mechanical stress, results in its constant interaction with various inflammatory mediators and causes severe chondritis and synovitis. In order to break this positive amplification loop, inhibiting the elevated NO levels might be a good approach.

However, in the presence of mature iNOS inhibitors, it might also be a thorny proposition to treat OA through this route due to various reasons. First, the increased dose of inhibitors or duration of treatment will unavoidably inhibit eNOS and nNOS due to the capped selectivity, which can affect the physiological regulation of vascular tone and blood pressure regulation and might lead to complications, such as hypertension [141]. Secondly, most OA patients might not start treatment until the disease has progressed to a certain extent, such as when severe pain occurs; the radiographic performance is often already obvious at this time. The altered mechanics in even radiographic “mild-to-moderate” OA are of such overwhelming significance that the current pharmacological agents might not effectively modify the outcome [142].

Nano-targeted drug delivery systems might be one of the future research directions that might significantly reduce related adverse reactions through precision drug delivery. In addition, it is a desirable prospect if there is the possibility to reverse even mildly altered mechanics by developing new pharmacological agents but not a prosthetic replacement.

## 7. Conclusions

The biological effects of NO are highly complex in human cartilage and thus in OA. The roles of NO are associated with various factors, such as IL-1β, TNF-α, and PGE_2_, and multiple signaling pathways, such as NF-κB activation. NO, thus, is a part of the following processes: regulating normal bone cell function and bone remodeling, responding to mechanical stimulation in bone, mediating the effects of sex hormones, inducing catabolic effects, and being an anti-inflammatory or pro-inflammatory factor in bone. Therefore, NO might be a new target for OA treatment. However, the effects of NO on normal and pathological bone and joint functions require further clarification using animal and cell studies. Additional clinical research on the applications of NO inhibitors in OA is also warranted.

## Figures and Tables

**Figure 1 molecules-28-01683-f001:**
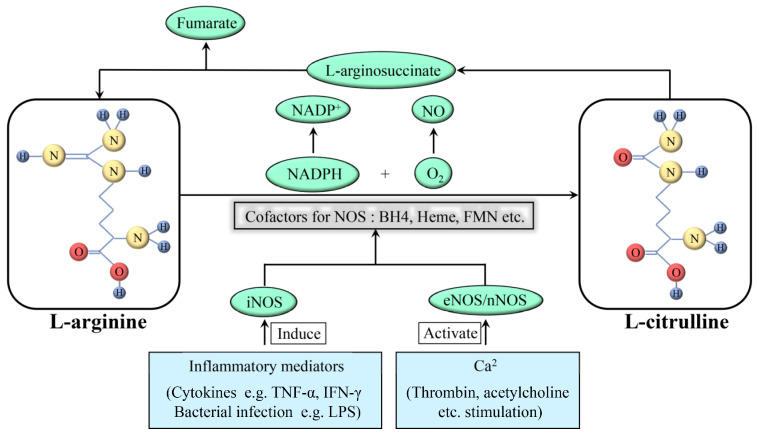
Formation and synthesis of NO. NAPDH: nicotinamide adenine dinucleotide phosphate; NAPD^+^: oxidation form of NAPDH; NO: nitric oxide; O_2_: oxygen; iNOS: inducible nitric oxide synthase; eNOS: endothelial nitric oxide synthase; nNOS: neuronal nitric oxide synthase; TNF: tumor necrosis factor; IFN: interferon; LPS: lipopolysaccharide.

**Figure 2 molecules-28-01683-f002:**
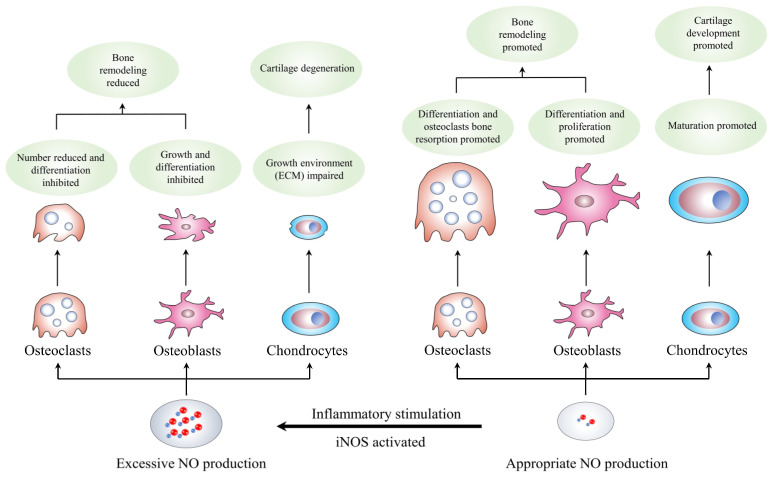
Formation and synthesis of NO. NO: nitric oxide; iNOS: inducible nitric oxide synthase; ECM: extracellular matrix.

**Figure 3 molecules-28-01683-f003:**
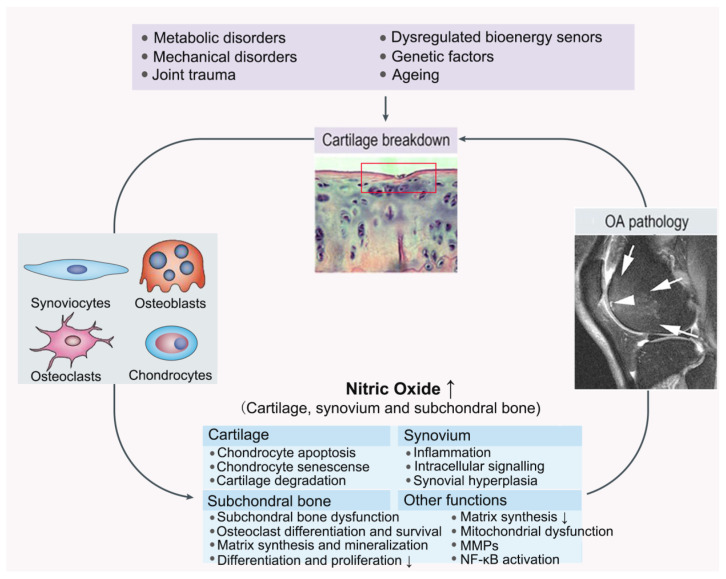
Joint damage caused through NO pathway in OA. OA: osteoarthritis; MMPs: matrix metalloproteinases; NF-κB: nuclear factor kappa-B.

**Figure 4 molecules-28-01683-f004:**
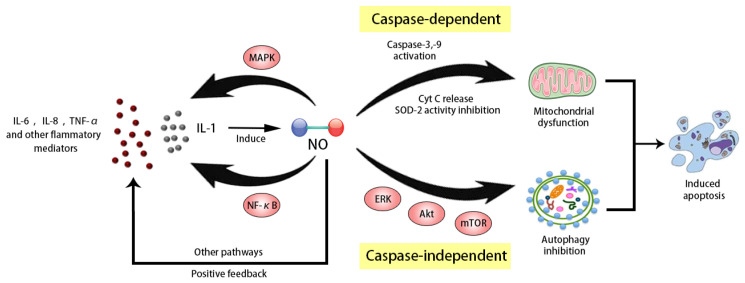
Mechanism of NO inducing apoptosis. IL: interleukin; TNF: tumor necrosis factor; MAPK: mitogen-activated protein kinase; NF-κB: nuclear factor kappa-B; NO: nitric oxide; caspase: cysteinyl aspartate specific proteinase; Cyt: cytochrome; SOD: superoxide dismutase; ERK: extracellular regulated protein kinases; Akt: protein kinase B; mTOR: mammalian target of rapamycin.

**Figure 5 molecules-28-01683-f005:**
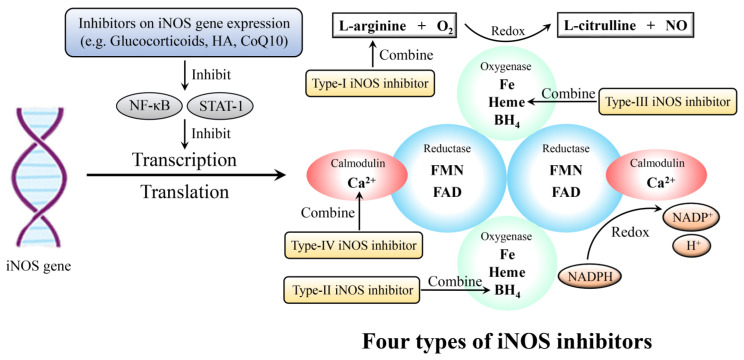
Inhibition to NO/NOS system. iNOS gene expression (transcription and translation) can be inhibited through NF-κΒ and STAT-1 signaling pathway. iNOS: inducible nitric oxide synthase; HA: hyaluronic acid; CoQ10: coenzymeQ10; NF-κB: nuclear factor kappa-B; STAT: signal transducer and activator of transcription; O_2_: oxygen; NO: nitric oxide; BH_4_: tetrahydrobiopterin; FMN: flavin mononucleotide; FAD: flavin adenine dinucleotide; NAPDH: nicotinamide adenine dinucleotide phosphate; NAPD^+^: oxidation form of NAPDH.

**Table 1 molecules-28-01683-t001:** Possible effect of NO on different type of cells and its related mechanisms.

Targeted-Cell	Type of Study	NO Concentration	Related Mechanisms	Effect of NO	Reference
Osteoclasts	In vivo and in vitro	High	Reduces the number of osteoclasts and inhibits its spread	Inhibits mature osteoclasts bone resorption	Kalyanaraman et al., 2017 [20]
Osteoclasts	In vitro	High	Induced by inhibited cGMP-degrading activity of PDE	Inhibits precellular osteoclasts differentiation	Amano et al., 2019 [21]
Osteoclasts	In vitro	Low	Induced by RANKL and produces downstream molecular 8-nitro-cGMP	Promotes osteoclasts differentiation	Kaneko et al., 2018 [22]
Osteoclasts	In vitro	Low	Stimulated by cytokines and other mediators such as PG	Promotes osteoclasts bone resorption	Mentaverri et al., 2003 [23]
Osteoblasts	In vitro	Low	Induces osteoblast differentiation factor (Cbfa1) expression	Promotes osteoblasts differentiation	Gloria et al., 2020 [24]
Osteoblasts	In vivo and in vitro	Low	Activates Src, Erk-1/2 and Akt signaling pathway through sGC and PKG2	Promotes osteoblasts proliferation and anti-apoptotic effects	Cepeda et al., 2020 [25] Ramdani et al., 2018 [26]
Osteoblasts	In vivo and in vitro	Low	Stimulated by estrogen	Promotes osteoblasts growth and development	Gerbarg et al., 2016 [27] Crescitelli et al., 2019 [28]
Osteoblasts	In vitro	Low	Responses to mechanical stimulation	Promotes osteoblasts proliferation and survival	Wittkowske et al., 2016 [29] Maycas et al., 2017 [30]
Chondrocytes	In vitro	High	Induces caspase expression upregulation	Induces chondrocytes apoptosis	Poderoso et al., 2019 [31] Kamm et al., 2019 [32]
Chondrocytes	In vivo and in vitro	High	Stimulated by inflammatory mediators	Affects numerous biomolecular processes in chondrocytes	Wojdasiewicz et al., 2014 [33]
Chondrocytes	In vitro	Low	Stimulated chondrocytes hypertrophy and increased the expression of alkaline phosphatase and type X collagen	Promotes chondrocytes maturation	Teixeira et al., 2005 [34] Drissi et al., 2005 [35]

**Table 3 molecules-28-01683-t003:** Part of the important pharmacotherapy agents of iNOS inhibition.

Author	Type of Study	Model	Agent	Dose	Method	Outcome	Conclusion
Yu et al., 2020 [100]	In vitro	New Zealand White rabbits	N-Monomethyl-L-arginine (L-NMMA)	Not mentioned	L-NMMA on inhibiting NO to accelerate the influence of simvastatin	L-NMMA inhibited NO and COX-2 production and NF-κB activation	L-NMMA enhanced the blocking effect of simvastatin on NF-κB activation by inhibiting NO production
Lee et al., 2012 [101]	In vitro	New Zealand White rabbits	L-NMMA	0.5 mM/ 24 h	L-NMMA on chondrocyte apoptosis	L-NMMA inhibited NO production and NF-kB binding activity	L-NMMA blocks PCB-initiated apoptosis effect
Eitner et al., 2021 [102]	In vitro	Human end-stage knee OA chondrocytes	N-Iminoethyl-L-lysine (L-NIL)	1, 10, or 20 µM/48 h	L-NIL on preventing release of NO, IL-6, PGE_2_, and iNOS	L-NIL prevented NO release and mitochondrial dysfunction	L-NIL improves the impairment of mitochondrial respiration
Bentz et al., 2012 [103]	In vitro	Human OA chondrocytes	L-NIL	0-20 µM/ 24 h	L-NIL on chondrocyte oxidative stress, apoptosis, inflammation, and catabolism	L-NIL stifled NO release, iNOS activity, nitrated proteins, and HNE generation and restored both HNE and GSTA4-4 levels	L-NIL prevents LPO process and ROS production and attenuates cell death, inflammation, and catabolism
Castro et al., 2006 [104]	In vivo	OA rats induced by ligament transection surgery	NG-nitro-L-arginine methyl ester (L-NAME)	30 mg/kg/bid	L-NAME on joint pain, cell influx, nitrite levels and iNOS expression	L-NAME reduced the time of rats’ right hind paw fails to touch the surface while walking	Prophylactic L-NAME can reduce joint pain
Järvinen et al., 2008 [105]	In vitro	Cartilage tissue from OA patients	1400W	1 mM/120 h	1400W on production of inflammatory mediators	Treatment with 1400W enhanced the production of anti-catabolic IL-10 and reduced MMP-10	The inhibiting effects of 1400W may point to its anti-inflammatory mechanisms for OA
Graverand et al., 2013 [106]	Clinical human study	Kellgren and Lawrence Grade (KLG) 2 or 3 knee OA patients	SD-6010	50 or 200 mg/day	A 2-year multicenter RCT of SD-6010 in patients with symptomatic knee OA	In KLG2 patients, JSN after 48 weeks was lower with SD-6010 50 mg/day versus placebo. No improvement in KLG3 patients	SD-6010 may become effective only in “mild to moderate” OA patients, but it cannot slow the rate of JSN
Ma et al., 2020 [107]	In vitro	IL-1β induced Sprague-Dawley (SD) rat chondrocytes	Aminoguanidine (AG)	0.3, 1 or 3 mM/24,48 or 72 h	AG on COX-2, iNOS, phosphorylated (p)-p65 and NF-κB translocation	AG downregulated iNOS and COX-2 expression by blocking the NF-κB signaling pathway	AG may protect chondrocytes and serve as a potential therapeutic for OA
Lee et al., 2013 [108]	In vivo	MIA-induced Wistar rat OA model	Coenzyme Q10 (CoQ10)	100 mg/kg/qd	CoQ10 on inflammatory mediators production and cartilage degradation	CoQ10 had anti-nociceptive effect and attenuated cartilage degeneration in rat OA model	CoQ10 exerts a therapeutic effect of OA by inhibiting inflammation
Park et al., 2018 [109]	In vitro	SW1353 cells and SD rats	Ethanol extract of sargassum serratifolium (EESS)	Extract that hard to measure precise concentrations	EESS on inflammatory mediators production and signaling pathways activation	EESS blocked ROS generation, attenuated NO production, and inhibited MAPK and PI3K/Akt pathways	EESS may have the potential chondroprotection in the prevention and treatment of OA
Khan et al., 2017 [110]	In vitro	IL-1β-stimulated human OA chondrocytes and cartilage explants	Wogonin	10–50 µM/24 h	Wogonin on inflammatory mediators production and MMPs, s-GAG and COL2A1 levels	Wogonin mediated Nrf2/ARE pathways, inhibited matrix degradation and suppressed the expression and production of COX-2 and iNOS	Wogonin exert chondro- and cartilage protection through the suppression of key molecular events
Yan et al., 2020 [111]	In vivo and in vitro	C57BL/6 wild-type (WT) rats	Myricitrin (Myr)	0-100µM/24h *in vitro*, dose in vivo not mentioned	Myr on inflammatory mediators production and signaling pathway	Myr suppressed the NF-κB and MAPK signaling pathways and decreased OARSI scores in OA rat models.	Myr may have therapeutic potential in the treatment of OA

## Data Availability

Not applicable.

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
