# Peer review of "Connection between Osteoarthritis and Nitric Oxide: From Pathophysiology to Therapeutic Target"

_molecules, 2023, doi:10.3390/molecules28041683_

Round 1
Reviewer 1 Report
The review by Zhou and coworkers provides a comprehensive insight into the role of nitric oxide in the pathogenesis of osteoarthritis and also discusses relevant literature with regard to the therapeutic strategies that might be useful in targeting the NO to ameliorate the degenerative potential in OA patients. The figures are quite attractive and self-explanatory for the audience. The content of the manuscript reflects a nice piece of literature. The manuscript would certainly be well received by the peers who are working in the areas of NO cell signaling or OA pathogenesis models. As minor issue some English language revision would be required, for example:
Abstract: Line 12: Replace "gas" with "gaseous"
Line 12: Please insert "is" after "that"
Line 14: Please replace "adjustment" with "regulation"
Introduction: Line 29 : Please replace "molecular" with "intracellular signaling molecule"
Line 38: Please replace "epidemiology" with "epidemiological"
Line 43: Please replace " societal" with "social"
So kindly have the whole manuscript screened for such better replacements.
Author Response
Dear Ms. Jasna Sotirovic (Editor) and the Reviewers:
Thanks both the Editor and the Reviewers for your kind support and hard work for the review to our manuscript entitled “Connection between osteoarthritis and nitric oxide: from pathophysiology to therapeutic target” (molecules-2189840). We would like to thank the reviewers for their valuable, suggestive and helpful comments for us to improve our manuscript. We have carefully modified our manuscript as suggested by the reviewers, as well as in accordance with guidelines of the technical editor. Our manuscript has now been edited by a native English speaker. All changes have been marked in “Red” word in the revision. Our responses (point to point) to the comments raise by the reviewers are as follows.
Reviewers’ comments:
Reviewer #1:
The review by Zhou and coworkers provides a comprehensive insight into the role of nitric oxide in the pathogenesis of osteoarthritis and also discusses relevant literature with regard to the therapeutic strategies that might be useful in targeting the NO to ameliorate the degenerative potential in OA patients. The figures are quite attractive and self-explanatory for the audience. The content of the manuscript reflects a nice piece of literature. The manuscript would certainly be well received by the peers who are working in the areas of NO cell signaling or OA pathogenesis models. As minor issue some English language revision would be required.
Response: Thanks sincerely for your view and comments, and we are truly glad to have your evaluation. The issues that you pointed out have been revised as follows.
Reviewer #1: Abstract: Line 12: Replace "gas" with "gaseous"
Response: Thanks for your careful view. We have corrected it in line 12 as “NO is a small gaseous molecule”.
Reviewer #1: Line 12: Please insert "is" after "that"
Response: Thanks for your careful view. We have corrected it in line 12 as “which is widely distributed in the human body”.
Reviewer #1: Line 14: Please replace "adjustment" with "regulation"
Response: Thanks for your careful view. We have corrected it in line 14 as “NO plays an important role in various physiological processes such as the regulation of blood volume and nerve conduction”.
Reviewer #1: Introduction: Line 29: Please replace "molecular" with "intracellular signaling molecule"
Response: Thanks for your careful view. We have corrected it in line 29 as “NO has been found to be a multipurpose intracellular signaling molecular”.
Reviewer #1: Line 38: Please replace "epidemiology" with "epidemiological"
Response: Thanks for your careful view. We have corrected it in line 40 as “According to epidemiological studies”.
Reviewer #1: Line 43: Please replace " societal" with "social"
Response: Thanks for your careful view. We have corrected it in line 45 as “thereby impairing the quality of life and increasing the economic and social disease burden”.

Reviewer 2 Report
The review by Jiang and colleagues summarizes the effects of NO on normal and OA joints and the possible new treatment strategies targeting the NO/NOS system. Besides the topic is of interest, several points should be addressed before publication.
1. The Authors should add a paragraph describing the signalling pathways activated by NO to have a complete view on the topic.
2. Lines 62-63: The dependence of NOS activity on BH4 should be acknowledge and taken into consideration for NOS uncoupling. In this light, a more comprehensive depiction of NOS enzymatic reaction should be provided.
Moreover, several concept must be clarified. For example:
-Line 48: “NO in the form of molecules such as sodium nitrite…”, sodium nitrate and the other cited compounds are not other forms of NO, but NO derivatives or NO derived molecules.
-Lines 56-57: “and its free electron on the N atom becomes rapidly oxidized”. An electron cannot be oxidized. Please correct.
-Lines153-154 the Authors wrote “Kaneko et al. found that 8-nitro-cGMP, the downstream molecular of NO and reactive oxygen species (ROS)”. However, 8-Nitro-cGMP is one of the cGMP derivatives that is formed when NO reacts with cGMP in the presence of ROS.
-Lines 181-182 the Authors wrote: “argininosuccinate lyase, an enzyme participating in synthesizing arginine and NO production”. However, argininosuccinate lyase converts argininosuccinate in arginine and fumarate with no net NO production.
In addition, several phrases should be re-written for clarity. Here are few examples:
Line 12 should be rephrased: “plays a key mediator in this process” should read: plays a role as key mediator in this process.
Line 39: “27 million–31 million people” should be “27 million out of 31 million people”.
Line 46: “contains” should be “depends on”
Similarly, Authors should check lines 142-143, 170-174, 201-203, 411-415, 460-462, 610-612, 649-650, 660-662
Lines 459-466 and lines 476-481 contain almost the same information, they could be merged.
Manuscript requires a thorough English revision.
The manuscript also contains several typos. For example:
Fig. 1 and Fig. 5: correct Heme
Line 103: respond instead of response.
Lane 108: stage instead of stages
Fig. 2 iNOS is underlined.
Line 645: duo instead of due
Please note that Acquisition of funding alone does not constitute authorship.
Author Response
Dear Ms. Jasna Sotirovic (Editor) and the Reviewers:
Thanks both the Editor and the Reviewers for your kind support and hard work for the review to our manuscript entitled “Connection between osteoarthritis and nitric oxide: from pathophysiology to therapeutic target” (molecules-2189840). We would like to thank the reviewers for their valuable, suggestive and helpful comments for us to improve our manuscript. We have carefully modified our manuscript as suggested by the reviewers, as well as in accordance with guidelines of the technical editor. Our manuscript has now been edited by a native English speaker. All changes have been marked in “Red” word in the revision. Our responses (point to point) to the comments raise by the reviewers are as follows.
Reviewer #2:
The review by Jiang and colleagues summarizes the effects of NO on normal and OA joints and the possible new treatment strategies targeting the NO/NOS system. Besides the topic is of interest, several points should be addressed before publication.
Response: Thanks sincerely for your very suggestive comment. The points have been revised as follows.
Reviewer #2: 1. The Authors should add a paragraph describing the signalling pathways activated by NO to have a complete view on the topic.
Response: Thanks for your valuable comments. We have added a new paragraph from line 666-677 as below.
“Current studies demonstrate that NO plays a critical role in activating various signaling pathways. Among these signaling pathways, NF-κB might be one of the most noteworthy. The activated NF-κB can promote the synthesis of proinflammatory factors, such as IL-1β, TNF-α, and iNOS, thereby aggravating the development of OA. In addi-tion, MAPK, STAT-1, and PI3K/Akt signaling pathways have similar functions. More-over, NO-activated ERK-Akt-mTOR signaling pathways can inhibit chondrocyte au-tophagy and promote apoptosis, and MMPs/TGF-β1/IGF-1 signaling pathways can in-crease matrix degradation, thereby causing cartilage tissue damage. On the other hand, the low NO concentrations can activate the RANKL signaling pathway, which produces downstream molecular 8-nitro-cGMP to promote osteoclasts differentiation, as well as Src-Erk-1/2-Akt signaling pathways, which can promote osteoblasts proliferation and has anti-apoptotic effects, thereby enhancing bone remodeling.”
Reviewer #2: 2. Lines 62-63: The dependence of NOS activity on BH4 should be acknowledge and taken into consideration for NOS uncoupling. In this light, a more comprehensive depiction of NOS enzymatic reaction should be provided.
Response: Thanks for your valuable comments. We have added extra description of NOS enzymatic reaction in line 64-70 and line 76-78 as below.
“NOS produces NO as a byproduct while oxidizing L-arginine to L-citrulline, using oxygen and nicotinamide adenine dinucleotide phosphate (NADPH) as substrates. In addition, this reaction requires flavin adenine dinucleotide (FAD), flavin mononucleotide (FMN), and (6R-)5,6,7,8-tetrahydro-L-biopterin (BH4) as cofactors. Among these cofactors, FAD and FMN are responsible for the transfer of electrons from NADPH to heme, and BH4 can bind to the oxygenase domain of NOS as an essential cofactor for substrates.”
“In some specific cases, such as oxidation of BH4 or depletion of L-arginine, eNOS can transform from a NO-producing enzyme to an enzyme that generates superoxide anion (O2-·), which is called NOS uncoupling”
Reviewer #2: Moreover, several concept must be clarified. For example:
-Line 48: “NO in the form of molecules such as sodium nitrite…”, sodium nitrate and the other cited compounds are not other forms of NO, but NO derivatives or NO derived molecules.
Response: Thanks for your suggestive comments. We have revised this description in line 50-52 as below.
“NO is found in the synovial fluid, serum albumin, and urine of OA patients in the form of NO-derived molecules such as sodium nitrite, fluorophenyltryptophan, and N-methyl-L-arginine”
Reviewer #2: -Lines 56-57: “and its free electron on the N atom becomes rapidly oxidized”. An electron cannot be oxidized. Please correct.
Response: Thanks for your suggestive comments. We have revised this description in line 58-59 as below.
“NO is an unstable, uncharged free radical, and the N atom in NO with free electrons gets rapidly oxidized.”
Reviewer #2: -Lines 153-154 the Authors wrote “Kaneko et al. found that 8-nitro-cGMP, the downstream molecular of NO and reactive oxygen species (ROS)”. However, 8-Nitro-cGMP is one of the cGMP derivatives that is formed when NO reacts with cGMP in the presence of ROS.
Response: Thanks for your suggestive comments. We have revised this description in line 163-165 as below.
“Kaneko et al. found that the 8-nitro-cGMP, which is a NO derivate that is formed when NO reacts with cGMP in the presence of reactive oxygen species (ROS), could promote RANKL-induced osteoclast differentiation”
Reviewer #2: -Lines 181-182 the Authors wrote: “argininosuccinate lyase, an enzyme participating in synthesizing arginine and NO production”. However, argininosuccinate lyase converts argininosuccinate in arginine and fumarate with no net NO production.
Response: Thanks for your suggestive comments. We have revised this description in line 190-193 as below.
“Recent studies on human somatic cells and mouse models showed that the osteoblasts lacked argininosuccinate lyase, an enzyme participating in the synthesis of arginine and contributing to NO production, which resulted in low NO production and failure to differentiate”
Reviewer #2: In addition, several phrases should be re-written for clarity. Here are few examples:
Line 12 should be rephrased: “plays a key mediator in this process” should read: plays a role as key mediator in this process.
Response: Thanks for your careful view and suggestive comments. We have revised this description in line 11-12 as below.
“Among the factors, nitric oxide (NO) plays a key role in mediating this process.”
Reviewer #2: -Line 39: “27 million–31 million people” should be “27 million out of 31 million people”.
Response: Thanks for your careful view and suggestive comments. After consideration, we have edited the sentence to clarify your concern in line 39-41 as below.
“According to epidemiological studies, one in every eight people in the United States (27-31 million people among the total US population) are affected by symptomatic OA”
Reviewer #2: -Line 46: “contains” should be “depends on”
Response: Thanks for your careful view and suggestive comments. We have revised this sentence in line 48-50 as below.
“Pathologically, the development of OA depends on chronic inflammatory processes including immune cell infiltration and the release of cytokine and metalloproteinase into the joints”
Reviewer #2: Similarly, Authors should check lines 142-143, 170-174, 201-203, 411-415, 460-462, 610-612, 649-650, 660-662
Response: Thanks sincerely for your careful view and valuable comments. We have checked these sentences carefully and edited them after deliberate consideration as below.
Line 151-153: “In this pathway, the effects of NO are cGMP-independent, and are different from the common pathway in most other systems.”
Line 181-186: “In other words, the high NO concentration can reduce the number of osteoclasts and inhibit their differentiation, demonstrating inhibitory effects on osteoclast bone resorption. On the other hand, the low NO concentration, mainly produced by cNOS, can promote the differentiation and survival of osteoclasts. Thus, the osteoclasts produced in nNOS-deficient individuals can lead to poor function. However, these conclusions are based on only in vivo and in vitro studies and lack the support of clinical data.”
Line 213-215: “In addition, based on estrogen stimulation, the moderate cNOS expression in osteoblasts can produce NO, which has a substantial role in the growth and development of osteoblasts as well as cytokine production”
Line 428-434: “Pathologically, the release of inflammatory molecules such as IL-1β and IL-6 aggravate inflammatory reactions and cartilage degeneration and cause structural changes, such as osteophyte growth, meniscal injury, and synovitis; thus, they become a direct reason for pain. In this process, NO causes a sustained release of inflammatory factors and is involved in the disease itself. Venkanna et al. conducted a study on S-methylisothiourea (SMT, an iNOS inhibitor) to investigate its role in relieving pain and inflammation.”
Line 479-486: “Currently, there are four types of iNOS inhibitors, which are the most widely known and are classified based on the inhibitor-binding sites on the NOS. The first type combines with the arginine site, part of which are reaction-based since they require an active enzyme and NADPH for complete inhibition”
Line 632-634: “In the animal group given the combination of USMSCs and GO, the NO contents decreased significantly in serum and articular fluid. GO granular lubricant might promote the therapeutic effects of UCMSCs as scaffold-carrying material”
Line 685-688: “However, in the presence of mature iNOS inhibitors, it might be also a thorny proposition to treat OA through this route due to various reasons. First, the increased dose of inhibitors or duration of treatment will unavoidably inhibit eNOS and nNOS due to the capped selectivity”
Line 697-699: “In addition, it is a desirable prospect if there is the possibility to reverse even mildly altered mechanics by developing new pharmacological agents but not a prosthetic replacement”
Reviewer #2: Lines 459-466 and lines 476-481 contain almost the same information, they could be merged.
Response: Thanks for your valuable comments. We have checked the information of these two paragraphs carefully and merged them in Line 479-489 as below.
“iNOS inhibitors can weaken iNOS activity. Currently, there are four types of iNOS inhibitors, which are the most widely known and are classified based on the inhibitor-binding sites on the NOS. The first type combines with the arginine site, part of which are reaction-based since they require an active enzyme and NADPH for complete inhibition; most inhibitors belong to the first type. The second type includes a set of compounds, which mimic the tetrahydrobiopterin cofactor. The third type interacts directly with heme, and some of them can bind to the heme of the enzyme monomer and prevent the formation of the active dimer; various anti-fungal imidazoles have been shown to inhibit NOS activity via this mechanism. The fourth type interacts with either calmodulin or falvine cofactors; however, this type of inhibitors is not considered promising means for selective iNOS inhibition”
Reviewer #2: Manuscript requires a thorough English revision.
Response: Thanks sincerely for your suggestive comments. Our manuscript has now been edited by a native English speaker. All changes have been marked in “Red” word in the revision.
Reviewer #2: The manuscript also contains several typos. For example:
Fig. 1 and Fig. 5: correct Heme
Response: Thanks for your careful view. The typos have now been corrected as below.
Reviewer #2: -Line 103: respond instead of response.
Response: Thanks for your careful view. The typos have now been corrected in Line 113 as “the stimulation of resorption in response to bone formation and the generation of new bone after bone degradation”.
Reviewer #2: -Line 108: stage instead of stages
Response: Thanks for your careful view. The typos have now been corrected in Line 117 as “paracrine signals harmonize human bone remodeling based on the adjustment of these cells in different stages”.
Reviewer #2: Fig. 2 iNOS is underlined.
Response: Thanks for your careful view. The typos have now been corrected as below.
Reviewer #2: - Line 645: duo instead of due
Response: Thanks for your careful view. The typos have now been corrected in Line 681 as “The pathological increase in NO contents due to stimulatory factors”.

Round 2
Reviewer 2 Report
The MS is now suitable for publication